# The role of fluctuations in quantum and classical time crystals

Toni L. Heugel,[1] Alexander Eichler,[2] R. Chitra,[1] and Oded Zilberberg[3]

[1]*Institute for Theoretical Physics, ETH Zürich, CH-8093 Zürich, Switzerland.*
[2]*Laboratory for Solid State Physics, ETH Zürich, CH-8093 Zürich, Switzerland.*
[3]*Department of Physics, University of Konstanz, 78464 Konstanz, Germany.*
(Dated: March 27, 2023)

Discrete time crystals (DTCs) are a many-body state of matter whose dynamics are slower than the forces acting on it. The same is true for classical systems with period-doubling bifurcations. Hence, the question naturally arises what differentiates classical from quantum DTCs. Here, we analyze a variant of the Bose-Hubbard model, which describes a plethora of physical phenomena and has both a classical and a quantum time-crystalline limit. We study the role of fluctuations on the stability of the system and find no distinction between quantum and classical DTCs. This allows us to probe the fluctuations in an experiment using two strongly coupled parametric resonators subject to classical noise.

## I. INTRODUCTION

The study of many-body periodically driven systems has experienced a boost in activity over recent years with the advent of discrete time crystals (DTCs) [1–22]. These are systems that respond to an external drive (at frequency $\omega_G$) and fulfill the following three criteria: (i) The period of the response is subharmonic at an integer multiple of that of the drive, or equivalently, its response frequency $\omega$ is an integer fraction of $\omega_G$, $\omega = \omega_G/n$ with $n$ integer; (ii) in its own rotating frame at $\omega$, the system appears to be stationary and exhibits long-time stability. In particular, no decay to a different state should occur when the system size approaches the thermodynamic limit; and (iii) The system exhibits sufficiently long-ranged correlations.

Much discussion is currently focused on the necessary and sufficient conditions to fulfill the three criteria listed above [23–28]. For example, discrete time-translation symmetry breaking (DTTSB) leading to (i) has long been known as "period doubling" in the field of nonlinear dynamics and is, for instance, a prominent phenomenon in Kerr Parametric Oscillators (KPOs) [29–35]. In this context, the many-body aspect (iii) naturally emerges from the existence of normal modes for both weak and strong coupling between local degrees of freedom [36]. Criterion (ii) received much attention in the context of closed quantum systems, where many-body localization and perfect quantum coherence are postulated to give rise to fully quantum DTCs [37]. The underlying assumption is that there are multiple states that perform DTC but remain decoupled from the rest of the system under the time-dependent drive and in the thermodynamic limit. This leads to the question whether many-body localized states exist under Floquet drives, which is beyond the scope of this work.

Any experimentally accessible system is finite and inherently open to the environment to some degree. One may therefore question its ability to maintain criterion (ii) in the presence of dissipation and decoherence [16, 27, 36, 38–43]. Specifically, examining the role played by fluctuations leads to a very basic question: in what sense does a realistic quantum DTC differ from a classical one in an open system? This discussion is intrinsically linked to the degree of coherence that can be achieved in any driven phase of matter.

In this paper, we answer the latter question by showing that the key features characterizing an open quantum DTC can be recovered within a purely classical and dissipative setting. We start from a Bose-Hubbard variant of a driven quantum many-body system [11, 44–46] and elucidate the role of dissipation and nonlinearity for its stationary behavior. We observe that the quantum many-body system forms normal modes [fulfilling (i) and (iii)] with DTC phases that mix over time through quantum fluctuations. This implies that criterion (ii) is only fulfilled for short and intermediate times [3, 47, 48], which we will refer to as the DTC lifetime in the following. We then compare the full quantum treatment with a mean-field picture, and investigate the quantum fluctuations of the DTC. Finally, we present a simple and classical experimental realization of a dissipative DTC and compare the quantum predictions to the measured results. Surprisingly, all the crucial aspects of the long-time behavior of the quantum model can be found in the classical experiment, demonstrating unambiguously that classical and quantum DTC share the same basic properties in the presence of dissipation. Our treatment highlights that condition (ii) is much easier to fulfill in a classical system, as its large amplitudes are more resilient to fluctuations.

## II. QUANTUM MANY-BODY TREATMENT

We begin by investigating DTTSB in a many-body closed quantum system, i.e., we identify its time-crystalline phases. Our starting point is a $N$-site network of interacting bosonic nodes subject to a homogeneous

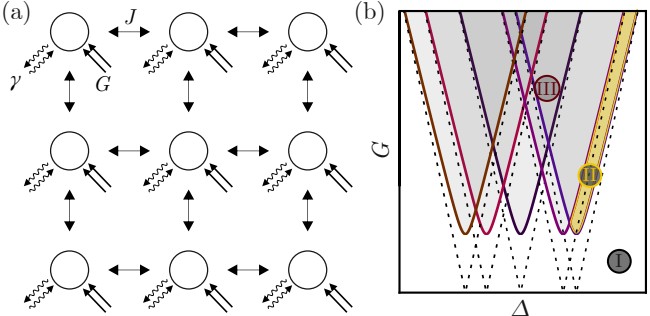

FIG. 1: (a) An example of a 2D graph of coupled parametric oscillators. They are nearest-neighbour coupled with amplitude $J$ and are subject to a homogeneous two-photon (parametric) drive G, and damping $\gamma$, cf. Eq. (1). (b) Schematic parametric instability lobes of normal modes: we approximate our coupled nonlinear resonators by the normal modes of the underlying harmonic oscillators, cf. Eqs. (3) and (4). These linear normal modes are unstable under the parametric drive whenever the drive amplitude exceeds a normal-mode parametric threshold, giving rise to a separate instability lobe for each normal mode [36, 49, 50]. Dashed lines are used for the closed system and solid lines with shaded regions for the open system (6).

two-photon drive

$$H(t) = \sum_j \hbar \Big[ \omega_j a_j^\dagger a_j + \frac{V_j}{12}(a_j^\dagger + a_j)^4 - e^{i\omega_G t}\frac{G_j}{2}a_j^\dagger a_j^\dagger$$
$$- e^{-i\omega_G t}\frac{G_j^*}{2}a_j\,a_j + \sum_{k\neq j}\frac{J_{jk}}{2}(a_j^\dagger + a_j)(a_k + a_k^\dagger)\Big], \quad (1)$$

where $a_j$ annihilates a bosonic particle (photon) on the $j^{\text{th}}$ site with respective eigenenergy $\omega_j$ and Duffing nonlinearity $V_j$, see Fig. 1(a) for a 2D example. The resonators are all-to-all coupled with amplitudes $J_{jk}$ and are driven with two-photon (squeezing, parametric) drives with amplitudes $G_j$ at a homogeneous drive frequency $\omega_G$.

Moving to a frame rotating at half the parametric two-photon driving frequency, i.e., by using the unitary transformation $e^{-i\sum_j a_j^\dagger a_j \omega_G t/2}$ and taking the rotating-wave approximation (RWA), we obtain an effective Hamiltonian of a $N$-site variant of the Bose-Hubbard model subject to two-photon squeezing terms, or equivalently $N$ coupled KPOs [46],

$$\bar{H} = \hbar \sum_j \Big[ -\Delta_j a_j^\dagger a_j + \frac{V_j}{2}a_j^\dagger a_j^\dagger a_j\,a_j - \frac{G_j}{2}a_j^\dagger a_j^\dagger$$
$$- \frac{G_j^*}{2}a_j\,a_j + \sum_{k\neq j}\frac{J_{jk}}{2}(a_j^\dagger a_k + a_j a_k^\dagger)\Big], \quad (2)$$

with linear detunings $\Delta_j = \omega_G/2 - \omega_j - V_j$. The RWA is equivalent to the lowest-order Floquet expansion, and relies on the fact that the corrections to the linear mode basis are nondegenerate and small [51]. Note that a more precise rotating approach exists [52], but the standard rotating frame suffices for our discussion here and appeals to a larger audience.

Coupled harmonic oscillators form normal modes. Therefore, by neglecting the parametric drive and Kerr nonlinearities, we can diagonalize Eq. (2) in a normal mode basis

$$H_{nm} = \sum_k \hbar \tilde{\Delta}_k b_k^\dagger b_k\,, \quad (3)$$

where $b_k$ denotes a linear combination of $a_j$ with detuning $\tilde{\Delta}_k$ of the $k^{\text{th}}$ normal mode. In this limit, our system appears to be composed of $N$ decoupled harmonic oscillator modes. We can apply the same normal-mode transformation also in the driven and nonlinear case. The role of the Kerr terms $V_j$ in the normal-mode basis is not trivial: the Kerr nonlinearities generate self-Kerr terms $V_k$ for each normal mode, and cross-normal-mode nonlinear couplings [50, 53]. Nevertheless, for weak nonlinearities and low-photon numbers, we can (to lowest order) neglect the cross terms and obtain approximately a system of $N$ decoupled normal modes with corresponding normal-mode self-Kerr terms.

The parametric drives in the normal basis give rise to eigenmode- and two-mode squeezing terms

$$H_{nm}^G = \hbar \sum_k \frac{\tilde{G}_{kk}}{2}b_k^\dagger b_k^\dagger + \hbar \sum_{l\neq k}\frac{\tilde{G}_{lk}}{2}b_l^\dagger b_k^\dagger + \text{h.c.}\,, \quad (4)$$

appearing as the terms containing $\tilde{G}_{kk}$ and $\tilde{G}_{lk}$, respectively. In our rotating picture, the former terms are important for DTTSB: they lead to instability lobes for the normal modes whenever the two-photon drive amplitude hits parametric resonance, i.e., whenever the system crosses the instability threshold $|\tilde{\Delta}_j| = |\tilde{G}_{jj}|$ of one of the normal modes, see Fig. 1(b) [54, 55]. Below this threshold, the two-photon drive does not excite the system and no time-crystalline phase is formed, see regime I in Fig. 1(b). Beyond this instability threshold, however, the modes undergo a DTTSB, where the system's normal-mode Kerr nonlinearity prevents the absorption of infinitely-many photons from the drive: with increasing amplitude, the mode's natural frequency shifts and becomes detuned from the fixed driving frequency. In this way, the nonlinearity prevents infinite growth of the mode and stabilizes the so-called time crystal [11, 36].

Stable solutions of a system of coupled KPOs were already found in Refs. [56, 57], and their different regimes were studied in detail in Refs. [36, 50, 58]. Here, we briefly review the regimes appearing in a diagram such as Fig. 1(b) for the convenience of the reader. In 'regime II', we observe that the $N$-coupled resonators are best described by our (approximate) construction of a parametrically driven (collective) normal mode that undergoes a spontaneous $\mathbb{Z}_2$ DTTSB, see Fig. 1(b) [36]. However, moving away from this regime by increasing the parametric drive or detuning, our approximations are no

longer valid, and we expect that the cross-Kerr coupling and two-mode squeezing terms couple the normal modes. In this 'regime III', the DTTSB involves a larger number of time-broken solutions, leading to a richer phase diagram [50]. Note that the latter regime involves the interplay of many normal modes that undergo DTTSB with a large overlap in the $\Delta$-$G$ parameter space [36]. Hence, there are two possible interpretations of many-body time crystals: on the one hand, one may be satisfied with a single collective mode that undergoes DTTSB, as in the fundamental mode of a massive mechanical resonator composed of many atoms [59, 60], or a row of strongly coupled pendula [50, 61]. On the other hand, one can demand that $N$ collective modes would undergo DTTSB and that their mutual interactions lead to new time-crystalline phases. The second case relies on weak coupling (or stronger driving) to allow for overlapping normal modes [36]. Both interpretations fulfill criteria (i) and (iii).

Now, we turn to discuss when criterion (ii) is fulfilled. As long as our approximate reasoning holds (regime II), each normal mode evolves coherently in its "cat-qubit" basis [62, 63] while being locked to half the frequency of the drive. This means that in this pure quantum limit, we will have a superposition of normal-mode DTCs. Depending on the initial conditions of the system and for carefully chosen $\Delta$ and $J$-values, we could then fulfill criterion (ii). These are the mathematical conditions postulated in many recent works for closed quantum DTCs [37]. There, many-body localization is assumed to suppress the cross-Kerr and two-mode squeezing terms (i.e., inter-mode interactions and two-photon drives) and keep the DTC states decoupled. These conditions can be broken in physical systems in the long-time limit. First, as discussed in Appendix B, as the localized modes are decoupled, each breaks the symmetry individually and in the thermodynamic system we will observe on average a very small symmetry breaking. Second, moving away from the decoupled pure quantum regime by relaxing our approximations, the inter-mode terms will perturbatively couple the different normal-mode DTCs. We expect this coupling to lead to loss of criterion (ii) through quantum oscillations that will manifest in the large coupled system limit (not discussed further in this work). Third, and most crucially, fluctuations play a decisive role in restoring a system's symmetry. In the following, we will devote our attention to the influence of fluctuations acting on our system, and to the long-time behavior resulting thereof.

Fluctuations enter through quantum mechanic's uncertainty principle and via coupling to an environment, leading to Langevin terms in the system's equation of motions. We now explicitly consider coupling to a zero-temperature dissipative environment resulting in weak single-photon loss terms at each site modeled as Lindblad dissipators

$$\mathcal{L}[\hat{O}]\rho = 2\hat{O}\rho\hat{O}^\dagger - \left\{\hat{O}^\dagger\hat{O}, \rho\right\},\qquad(5)$$

with associated rates $\gamma_j \ll \omega_0$. The time evolution of the system's density matrix $\rho$ is given by

$$\dot{\rho} = -\frac{i}{\hbar}[\bar{H}, \rho] + \sum_l \frac{\gamma}{2}\mathcal{L}[a_l]\rho,\qquad(6)$$

where dots indicate time derivatives and $\gamma_j \equiv \gamma$. In Eq. (5), the anti-commutator term corresponds to dissipation, while the other so-called *recycling* or *quantum jump* term encodes fluctuations and ensures the normalization $\mathrm{Tr}\rho \equiv 1$ of the system's density matrix. Equation (6) is the many-body extension of the quantum model used in Ref. [46], and establishes a quantum description of the classical many-body time crystal studied in Ref. [36].

Opening our system to weak dissipation channels does not impact the stability diagram in a radical way. The dissipation primarily results in a slight shift of the aforementioned instability lobes [55, 64], see the difference between dashed and solid lines in Fig. 1(b). Linear damping cannot compensate for the 'exponential' two-photon driving term [55, 62, 64]. Hence, we emphasize that the time-crystalline phases are still stabilized by the nonlinearity, and that there is no fundamental difference between dissipative and non-dissipative DTCs. Yet, as we shall now see, the bath-induced quantum fluctuations will mix DTCs in a finite-sized system over long enough timescales. This mixing process limits the duration in which criterion (ii) prevails even at zero temperature.

## III. EXAMPLE WITH $N = 2$

To illustrate the time-crystalline phases discussed above, we study in the following the case of $N = 2$. Whenever possible, we use a general indexing approach to illustrate the generality of the treatment for $N > 2$, albeit sometimes numerically challenging to solve. We numerically solve the Lindblad master equation (6) for stationary distributions using QuTiP [65] for the case of two identical oscillators ($\Delta_j = \Delta$, $V_j = V$, $G_j = G$, and $J_{jk} = J$ with $j \neq k$), see Fig. 2(a). Here, we expect in the linear limit to only have a single symmetric (S) and a single antisymmetric (A) normal mode, at detunings $\Delta = \pm J$ with associated parametric instabilities at $|G| \geq \sqrt{(\Delta - J)^2 + (\gamma/2)^2}$ and $|G| \geq \sqrt{(\Delta + J)^2 + (\gamma/2)^2}$, respectively, cf. solid lines in Fig. 2(b) and Ref. [50].

In Figs. 2(c)-(e), we present the joint probability distribution $\langle x_1, x_2|\rho_s|x_1, x_2\rangle$ obtained when measuring $(x_1, x_2) \equiv (a_1 + a_1^\dagger, a_2 + a_2^\dagger)/\sqrt{2}$ simultaneously on the stationary $\rho_s$. We observe "hot spots" where the distribution is high, with a finite width due to quantum fluctuations. In Fig. 2(c), representative for regime I, we see that the distribution below the instability threshold is centered around $x_1 = x_2 = 0$.

When the system is driven at $\omega_G$, we observe hot spots of distinct oscillation modes moving at $\omega_G/2$, which is the signature of a DTTSB. Specifically, moving beyond

the threshold of the symmetric normal mode, we obtain an example of regime II in Fig. 2(d): each hot spot is displaced, corresponding to a $\mathbb{Z}_2$ symmetry-breaking for the symmetric mode (S-state). A similar scenario is observed slightly above the threshold of the antisymmetric mode (A-state) [50]. By increasing the drive beyond the parametric instability threshold, we enter regime III and observe that additional time-broken phases appear in the stationary distribution of our system, dubbed mixed-symmetry states (M-states), see Fig. 2(e). Depending on the system parameters, we find various such mixtures of states [50].

The quantum fluctuations lead to mixing between the DTTSB regions in phase space. Thus, after long times, a finite-sized system approaches a "thermalized" distribution that mixes between different time-crystalline states, with a vanishing statistical mean amplitude. This implies that when the system is initialized around one of the hot spots, the time crystal phase will exist for a certain time before mixing by quantum fluctuations [62, 63]. Such fluctuation-activated mixing was studied for individual oscillators and for weakly coupled systems, see e.g. Ref. [11] and references therein. The related problem of quantum heating at zero temperature was addressed in Ref. [66]. Crucially, however, the statistical mixture is composed only of subharmonic states and the system can therefore only tunnel between time-crystalline phases akin to the closed system. The distributions shown in Fig. 2(d) and (e) are therefore involving only symmetry-broken contributions and can be distinguished from featureless thermal distributions, such as shown in Fig. 2(c), by correlation measurements. The study of the dynamics of tunneling events between the hot spots is an interesting topic [67–70] that can characterize the DTC lifetime, i.e., the intermediate time under which the system fulfills criterion (ii).

## IV. SEMICLASSICAL TREATMENT

It is instructive to separate the stationary distributions in Fig. 2 into stationary points (attractors, hot spots) $\alpha_j$ and small fluctuations $\delta a_j$ away from these points. To this end, we insert a mean-field ansatz $a_j = \alpha_j + \delta a_j$ to our model (2), with coherent states $\alpha_j \equiv \langle a_j \rangle$ and quasiparticle fluctuations $\delta a_j \equiv a_j - \langle a_j \rangle$. We characterize the steady states ($\dot{\alpha}_j = 0$) of the system by solving the resulting mean-field equations of motion [36, 53, 71, 72]

$$\dot{\alpha}_j = i(\Delta_j \alpha_j - V_j \alpha_j^* \alpha_j \alpha_j + G\alpha_j^* - \sum_{k \neq j} J_{jk}\alpha_k) - \frac{\gamma_j}{2}\alpha_j + \xi \,. \tag{7}$$

Note that this is equivalent to either employing a saddle-node approximation on a Keldysh action description of the system [53], or solving the corresponding classical system [50]. The noise enters Eq. (7) as an additional uncorrelated stochastic (Langevin) force $\xi = \Xi_{\alpha_{\mathrm{Re},j}} + i\Xi_{\alpha_{\mathrm{Im},j}}$, acting on the real and imaginary parts

of the coherent states with power spectral densities $\sigma^2 = \varsigma^2/2(\omega_G/2)^2$ [73–75].

The resulting semiclassical solutions are located at the hot spots of the projected probability distribution $\langle x_1, x_2|\rho_s|x_1, x_2 \rangle$, see Fig. 2. We conclude that the mean-field approach identifies the states that contribute most to the stationary density matrix $\rho_s$. In other words, we find that even with very few photons, the stationary density matrix that mixes between the DTTSB quantum states is located around specific hot spots in phase space, i.e., at the semiclassical coherent states of the driven system. This is at the root of the similarity between quantum and classical DTCs. Thus, we can characterize the phase diagram as a function of driving strength $G$ and detuning $\Delta$, cf. Refs. [50, 53] and Fig. 2(b). Nevertheless, so far we do not explain the uncertainty distribution around the hot spots nor how the system could ergodically explore the different hot spots in phase space.

*Fluctuations* – One hallmark of quantum mechanics is the inherent uncertainty accompanying all states in the form of quantum fluctuations. This uncertainty gives rise to the finite probability distribution around the hot spots, see Fig. 2. To account for these fluctuations, we consider the Hamiltonian

$$H_{\mathrm{fl}} = \hbar \sum_{kl} \Omega_{kl}(\alpha)\delta a_k^\dagger \delta a_l + \hbar \sum_{kl} S_{lk}(\alpha)\delta a_k^\dagger \delta a_l^\dagger + \mathrm{h.c.}\,, \tag{8}$$

where we neglect terms that are more than bilinear in $\delta a_j$. We have eigenfreqencies and couplings with the prefactors $\Omega_{kl}$ (for $k = l$ and $k \neq l$, respectively) and eigen- or cross-squeezing with prefactors $S_{kl}$ (for $k = l$ and $k \neq l$, respectively), all of which depend on the specific mean-field solution $\alpha$. Note that a similar treatment of fluctuations is possible for open systems [53, 71, 76, 77]. In both closed and open cases, the resulting fluctuations describe the dynamics of deviations away from the mean-field solution [53, 77]. The time evolution of the fluctuations is governed by characteristic exponents $\mu_i$ that are obtained by diagonalizing the Hamiltonian or Liouvillian, respectively. Alternatively, they can be read from the poles of the Keldysh Green's functions [53]. The real and imaginary parts of each $\mu_i$ correspond to the typical frequency and decay rate of the fluctuations around a phase of the system, respectively. Note that the fluctuation-induced activation dynamics between phases is not captured by the bilinear description (8), and can be described, e.g., using an instanton analysis [67–70, 78].

So far, we considered quantum fluctuations in a system coupled to a bath at temperature $T = 0$, cf. Eq. (6) and Fig. 2. In the limit of high normal modes' amplitudes, the influence of these quantum fluctuations rapidly decreases concomitant with a suppression of activation, cf. Appendix A. This results in long lifetimes and the DTC remains coherent over a long duration. In the classical limit, significant fluctuations can still enter the system from a high-temperature bath. We refer to this as the thermal limit $k_B T \gg \hbar\omega_0$, where $k_B$ is Boltzmann's constant. The classical thermal fluctuations, which in con-

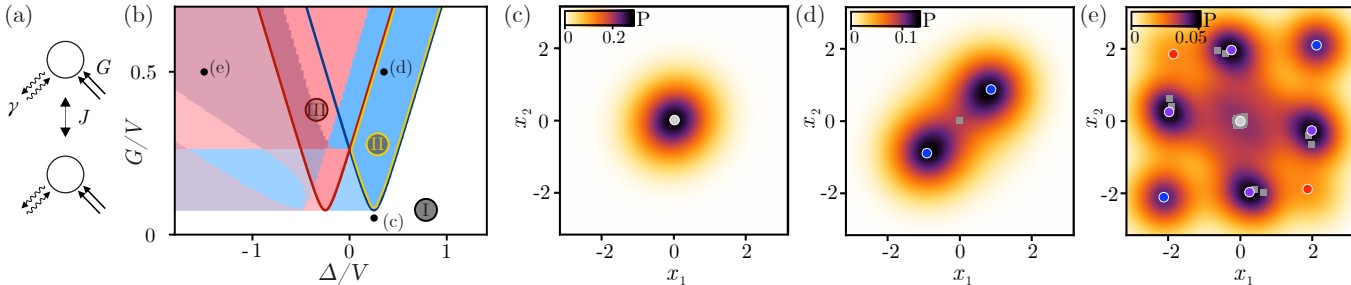

FIG. 2: (a) Network of two identical coupled parametric oscillators. (b) Calculated mean-field stability phase diagram, cf. Eq.(7). White: 0-state is stable; blue: only the S-state is stable; red: S and A-states are stable; purple: S and M-states are stable; dark red: S, A and M-states are stable; Brighter colors: also 0-state is stable. (I-III) indicate the below-threshold regime (I), the simple normal-mode regime (II), and the more complex partial overlapping regime (III) of the instability lobes (marked by blue and red solid lines). (c-e) Probability distribution of the quantum steady state $\langle x_1, x_2 | \rho | x_1, x_2 \rangle$ of measuring $(x_1, x_2)$ simultaneously. Circles indicate the stable mean field steady state solutions and squares indicate the unstable ones, cf. Eq. (7). Their colors indicate their symmetry: blue means symmetric, red means antisymmetric, purple means mixed symmetry and light-gray means 0-amplitude state. (c) Only the 0-amplitude state is occupied. (d) The symmetric states are populated, as they are the only ones with stable mean field solutions. (e) Many states are populated, the M-states are the most probable ones. Parameters are $\gamma/V = 0.1$, $J/V = -0.25$, and detuning and driving strength as shown in (b).

trast to quantum fluctuations can be tuned, should generate similar long-time phenomena with the same characteristic exponents as quantum fluctuations. To test this, we now move to an experimental realization of two coupled parametric oscillators in the presence of classical noise.

## V. EXPERIMENTAL REALIZATION

We set out to test if the fluctuation spectra of a time crystal can be observed in a classical experiment. To this end, we study a system of electrical lumped-element RLC resonators that are capacitively coupled. Each resonator is composed of an inductance $L = 87\,\mu\mathrm{H}$ and a nonlinear capacitance $C \approx 20\,\mathrm{pF}$ in the form of a dc-biased varicap diode, see Fig. 3(a). Finite damping enters the system through the electrical resistance, which mostly stems from the contribution of the coil wire. Each resonator is inductively connected to the lock-in amplifier's input and output ports via two auxiliary coils.

Under appropriate driving, each RLC resonator behaves as a KPO. The resonator network can therefore be modeled with the classical limit of Eq. (1) [79]. The parameter values of our resonators are (nearly) identical with eigenfrequencies $\omega_j \approx \omega_0 = 2\pi f_0 = 2\pi \times 2.603\,\mathrm{MHz}$, $V_j \approx V_0 = 2.56\,\mu\mathrm{Hz}$, and quality factors estimated as $Q_j \approx Q_0 = 233$, leading to homogeneous dissipation coefficients $\gamma_j \approx \gamma_0 = \omega_0/Q_0$. The resonators are driven with the same parametric driving strength $G_j \equiv G = \gamma_0 U_d/(2U_{th})$, where $U_d$ specifies the voltage applied to the driving coil, and $U_{th} = 1.95\,\mathrm{V}$ its threshold for parametric oscillation at $\Delta = 0$. In the following, we discuss experimental results obtained with two resonators with a linear coupling coefficient $J = 0.084\,\mathrm{MHz}$, placing them in a regime of moderately strong coupling [36, 50].

In previous work, we investigated the steady-state solutions of the coupled system in the $\Delta$-$G$ map [50]. We found a complex phase diagram of time-crystalline solutions with different symmetries. The theoretical analysis of this previous experiment, which is shown in Fig. 2(b), serves as an orientation map in the following discussion. Here, we are interested in the study of fluctuations around the stable solutions of Fig. 2(b). With $T \approx 300\,\mathrm{K}$, our resonator is deep in the classical limit $\hbar\omega_0 \ll k_B T$. This means that the thermal occupation of $n_{th} \approx 2.5 \times 10^6$ generates a sizeable sampling around the mean-field hot spots, overwhelming the influence of quantum fluctuations. This reiterates the truly classical nature of the physics we measure.

A natural source of classical fluctuations is provided by thermal Johnson noise. However, the room-temperature Johnson noise in our circuit is very small and hard to distinguish from the background noise of the detector (amplifier) noise. Instead, we apply electrical white noise $U_{\mathrm{noise}}$ with an approximately white power spectral density $S_n$. In this way, we mimic the effect of thermal noise at very large temperatures, and measure the resulting fluctuations around various states, where $\varsigma^2 = 0.0035\,\mathrm{Hz}^4/\mathrm{V}^2 S_n$ takes into account the signal incoupling efficiency and the rotating transformation performed in order to obtain Eq. (7).

## VI. EXPERIMENTAL RESULTS

Our experimental procedure is the following: we perform measurements as a function of the parametric driving frequency $2f_d = 2f_0 + \frac{\Delta}{\pi}$ at a constant driving amplitude $U_d \propto G$. At every value of $f_d$, we wait until a stationary oscillation is reached. This steady state corresponds to one of the time-crystalline phases in Fig. 2(b),

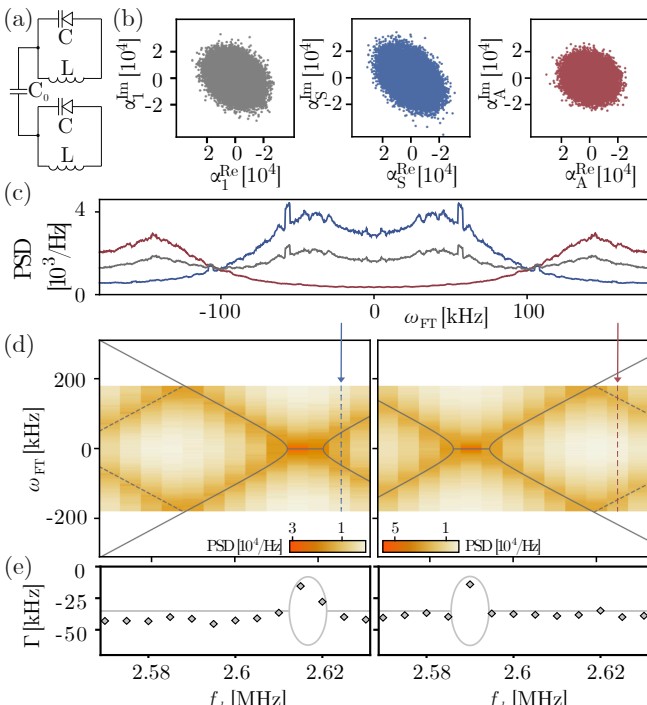

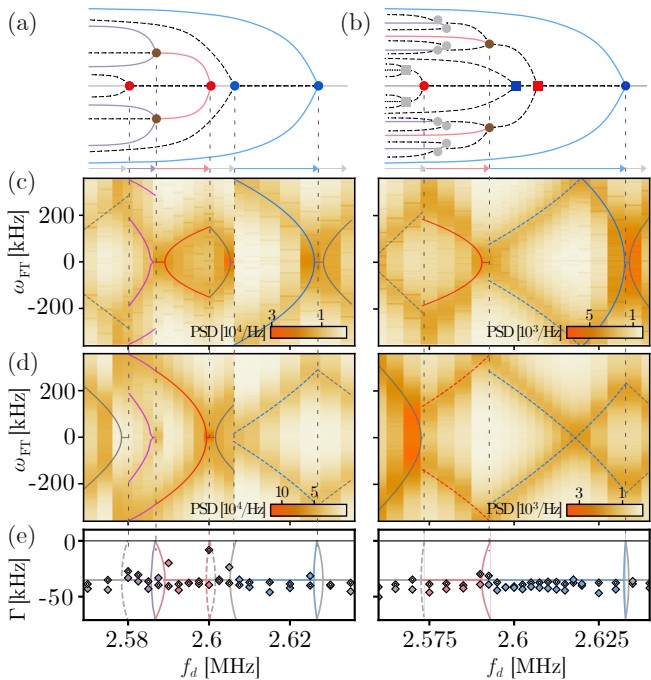

FIG. 3: (a) Experimental setup of two coupled parametric electric resonators. (b) Fluctuations around a stable state involving two coupled parametrons. Shown are the measured oscillations of oscillator 1 (gray), as well as the symmetric (blue) and antisymmetric coordinates (red) calculated from data of both resonators, cf. Eqs. (9) and (10). (c) Measured PSD of oscillator 1 (gray), and of the symmetric (blue) and antisymmetric (red) fluctuations. Fluctuation peaks at different frequencies become visible for the different symmetries. A moving average was used to decrease the point-to-point noise in the PSD. (d) PSD of the symmetric (left) and antisymmetric (right) fluctuations as a function of the driving frequency $f_d$. The driving strength used here is below threshold, i.e., no DTC appears and the mean amplitudes are zero. Grey lines show the calculated imaginary parts of the characteristic exponents $\mu$, cf. Appendix D. Aliasing appears due to the finite sampling rate [80, 81]. The affected values are indicated by dashed grey lines that are mirrored with respect to the maximum measured frequency. Red and blue dashed vertical lines and arrows mark the position of the measurements presented in (b) and (c). (e) Real parts of the characteristic exponents $\mu$, corresponding to damping; diamonds represent experimental data and lines the theory results. System parameters are $U_d = 1.5\,\mathrm{V}$, $S_n = 1.1 \times 10^{-9}\,\mathrm{V^2\,Hz^{-1}}$ and $f_d = 2.605\,\mathrm{MHz}$.

FIG. 4: (a) and (b): bifurcation tree of the experimental system comprising two coupled nonlinear resonators. Solid (dashed) lines indicate stable (unstable) DTC states for driving voltages of $U_d = 4\,\mathrm{V}$ in (a) and $U_d = 6\,\mathrm{V}$ in (b). Dots mark bifurcation points where several solutions emerge/vanish. The higher driving voltage in (b) results in a more complex bifurcation tree due to the larger overlap between the instability lobes, cf. Fig. 2(b). At each value of $f_d$, the experimental system probes one of the stable solutions, which we color code according to their stationary symmetry: grey (0-amplitude), blue (symmetric), red (anti-symmetric), and purple (mixed-symmetry). As in Fig. 3, we plot the PSD of the (c) symmetric and (d) anti-symmetric fluctuations around the stationary DTC states as a function of the driving frequency $f_d$, cf. Eqs. (9) and (10). Jumps in the realized DTC solutions are reflected as discontinuities in the measured spectra. We deal with aliasing as explained in the caption of Fig. 3. (e) Real parts of $\mu$, corresponding to damping; diamonds represent experimental data and lines the theory results. $S_n = 1.1 \times 10^{-9}\,\mathrm{V^2\,Hz^{-1}}$ and $f_d = 2.605\,\mathrm{MHz}$.

i.e., a mean-field solution with a certain amplitude and phase for each resonator. Next, we record the fluctuations of each resonator around this stable state in response to the combination of the parametric drive $U_d$ and the noise $U_{\mathrm{noise}}$. This procedure is akin to a pump-probe experiment, where the pump stabilizes a stationary state, and the probe is a white noise drive. We dub this procedure 'pump-noisy-probe'. To obtain statistically representative data, the measurement time $T_{\mathrm{rec}}$ must be much

longer than both the lock-in time constant and the resonator ringdown time.

For a single, isolated resonator, we can Fourier transform the measured quadrature displacements $\delta\alpha^{\mathrm{Re}} \equiv \mathrm{Re}\{\bar{\delta a}\}$ and $\delta\alpha^{\mathrm{Im}} \equiv \mathrm{Im}\{\bar{\delta a}\}$, where bars indicate semiclassical short-time-averaged values (on top of the stationary states). These excitations fluctuate around the mean values and are similar to thermal oscillations of a harmonic oscillator but around a displaced frame. The position and width of the peaks in the resulting spectrum allow us to extract the excitation frequency $\omega_{\mathrm{FT}}$ and decay rate $\Gamma$, see Appendix C.

For coupled resonators, the excitations away from the stable states are themselves coupled and form normal modes (excitation quasiparticles), see Fig. 3(b). As we

measure the response of both resonators individually, we can analyze the excitations in terms of their symmetric and antisymmetric normal-mode components, which are defined as

$$\delta\alpha_{S,A}^{\mathrm{Re}} = (\delta\alpha_1^{\mathrm{Re}} \pm \delta\alpha_2^{\mathrm{Re}})/\sqrt{2}\,, \qquad (9)$$

$$\delta\alpha_{S,A}^{\mathrm{Im}} = (\delta\alpha_1^{\mathrm{Im}} \pm \delta\alpha_2^{\mathrm{Im}})/\sqrt{2}\,. \qquad (10)$$

The advantage of this procedure is that it isolates the spectral components of the excitations, and allows us to track the individual resonances in the presence of aliasing. We emphasize that the symmetries of the excitations are independent of the symmetries of the selected stationary states. The former are observed as perturbations within the frame rotating at the frequency of the latter. For instance, the excitations around a symmetrical normal-mode steady state exhibit both symmetric and antisymmetric components. In general, these components appear at different frequencies $\omega_{\mathrm{FT}}$, which allows us to distinguish them in the Fourier transform of the corresponding coordinate, see Fig. 3(c).

As a starting point, we probe the excitations of the coupled system as a function of $f_d$ below the parametric threshold, i.e., at small driving $U_d$ where no time-crystalline phases exist. This corresponds to a horizontal cut through regime I, cf. Fig. 2(b). In Fig. 3(b), we display the excitations of resonator 1 in its own coordinates $\alpha^{\mathrm{Re}}$ and $\alpha^{\mathrm{Im}}$, as well as the symmetric and antisymmetric excitations of both resonators. Since we are below threshold, $\bar\alpha_i = 0$, and we can identify $\delta\alpha_i = \alpha_i$ in this case. For detunings $\Delta$ close to the symmetric instability lobe, we observe an elongated (squeezed) phase space distribution for $\alpha_S$, while $\alpha_A$ remains round [76]. The corresponding power spectral densities (PSD) are shown in Fig. 3(c), demonstrating the different frequencies of the symmetric and antisymmetric excitations. Note that both are far below the resonance frequency $f_0 = 2.603\,\mathrm{MHz}$.

In Fig. 3(d) and (e), we summarize the PSDs of the excitations measured at different frequencies, and compare them to the values calculated with our analytic prediction for the characteristic exponents $\mu_i$, cf. Appendix D. The predicted excitation frequencies and bandwidths match those of the measured PSDs. In particular, we observe regions where the excitations have a well-defined frequency and lifetime. At other points, the frequencies go to zero while the decay rate splits due to the phase-dependent parametric amplification [53, 82]. The eigenvalue diagram is similar to that of a damped harmonic resonator going from an underdamped to an overdamped motion at a so-called 'exceptional point' [53]. We note that multiple such scenarios emerge in our network.

For $U_d > U_{th}$, the driving is sufficiently strong to generate DTCs in a certain frequency range. In Fig. 4, we explore the excitations in this regime for two different values of $U_d$. The two sweeps represent cuts through the diagram in Fig. 2(b) below and above the point where the two instability lobes start to overlap, respectively. In general, the system possesses a multitude of stable states

with a complex bifurcation tree, see Fig. 4(a) and (b). The experiment samples one particular oscillation state at each frequency. Hence, the experimental sweeps follow trajectories along the bifurcation diagram, see Figs. 4(c) to (e). A jump between different states, typically at a bifurcation point, causes a jump in the measured excitation spectrum [83]. Several such jumps can be observed in our measurements. Again, all measured spectra can be modeled using the characteristic exponents $\mu_i$, in spite of the fact that our devices are deep in the classical regime. The main difference between the closed and open cases involves the finite lifetime of the excitations, see Fig. 4(e).

## VII. DISCUSSION AND OUTLOOK

Our results demonstrate that DTC physics has a clear connection to classical period-doubling bifurcations [29, 33] and is strongly impacted by fluctuations. Any finite-sized system includes fluctuations that lead to loss of coherence and restore symmetries over sufficiently long times [35]. In our concrete analysis of a DTC built from KPOs, we showcase that the long-time limits of both the quantum and classical limits stem from the same model description and exhibit the same mean-field order parameters as well as excitations. This finding entails that in our system, the criterion (ii) delineated as a requirement for DTCs is solely maintained within the DTC lifetime [3, 47, 48]. Interestingly, as massive mode populations provide resilience against fluctuation-induced activation between various attractors, systems with large amplitudes have diverging lifetimes. This condition is easy to fulfil in classical systems.

Our study relies on a specific model that maps to our classical experiment. Nonetheless, we emphasize that the physics seen should be applicable to a wide range of interacting systems, including DTCs realized with spins. For instance, using bosonization techniques, such systems map to similar models and phenomenology as studied here [53]. This leads us to wonder whether current realizations of quantum DTCs [1–3, 6, 19–21] can be interpreted as a manifestation of DTTSB whose mean-field behavior in the long-time limit is analogous to classical examples in the literature, e.g. Ref. [30]. Most importantly, we demonstrate how the excitations around the out-of-equilibrium quasi-stationary solutions of a KPO network can be measured and analyzed. The agreement between the experiment and the theory is quite remarkable and motivates similar studies in other DTC systems.

### Acknowledgments

This work received financial support from the Swiss National Science Foundation through grants (CR-SII5_177198/1) and (PP00P2_190078), and from the Deutsche Forschungsgemeinschaft (DFG) - project number 449653034. We thank Peter Märki and Žiga Nosan

for technical help.

## Appendix A: Classical system

The classical version of Eq. (1) corresponds to a coupled parametric oscillator network [Eq. (1)] and is described by the following equation of motion [70]:

$$\ddot{x}_j + \omega_j^2(1 - \lambda \cos \omega_G t)x_j + A_j x_j^3 + \sum_{k \neq j} J_{jk}\sqrt{\omega_j \omega_k}x_k , \quad (A1)$$

where the parametric driving is related to the two-photon drive via $\lambda = 4G/\omega_j$, and the Duffing-nonlinearity to the Kerr-nonlinearity via $A_j = 4\omega_j^2 V_j/3\hbar$. Note the explicit dependence on $\hbar$ in the latter relation because of the nonlinearity. In the classical limit $\hbar \to 0$, $A_j$ does not depend on $\hbar$. The description of the classical system can be simplified analogously to the quantum system by going into a rotating frame at $\omega_G/2$ and neglecting fast oscillating terms yielding equations corresponding to the mean-field equations for $\alpha_j$, Eq. (7) [36, 71, 72].

## Appendix B: Symmetry of cat states

The premise of quantum DTCs rests on the assumption that a large number of decoupled elements (be it two-level systems or oscillators) respond to an external drive simultaneously. Each of the elements will transition into a superposition of two symmetry-broken states, such as the two spin polarization states along a selected axis. In a network of KPO normal modes, as discussed in the main text, the two symmetry-broken states are coherent states $|\alpha\rangle$ with amplitudes

$$\alpha = \pm\sqrt{G/V} . \quad (B1)$$

Depending on the initial condition, the state that each KPO mode attains in response to the parametric drive is a superposition of such coherent states described by

$$\rho = c_+ |C_+\rangle \langle C_+| + c_- |C_-\rangle \langle C_-| . \quad (B2)$$

Here, the so-called cat states $|C_\pm\rangle$ are themselves superpositions of the underlying coherent states,

$$|C_\pm\rangle = c_{\pm\text{cat}} (|\alpha\rangle \pm |-\alpha\rangle) , \quad (B3)$$

and the $c_{\pm\text{cat}}$ are appropriate normalization factors [62]. It should be noted that these superpositions do not generally possess a broken symmetry.

Assuming that the individual modes (with index $i$) individually achieve maximal symmetry breaking with

$$\rho_i = |\pm\alpha_i\rangle \langle \pm\alpha_i| , \quad (B4)$$

the network in total will be formed by the sum

$$\rho_{\text{tot}} = \sum_i |\pm\alpha_i\rangle \langle \pm\alpha_i| . \quad (B5)$$

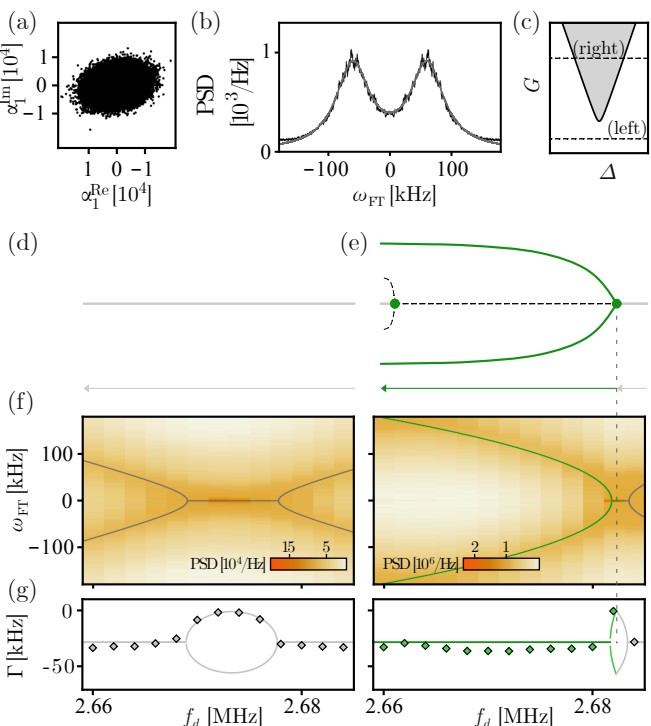

FIG. 5: Fluctuations around stable states in a single parametron. (a) Measured fluctuation around the origin for $U_d = 1.3\,\text{V}$, $S_n = 1.1 \times 10^{-9}\,\text{V}^2\,\text{Hz}^{-1}$ and $f_d = 2.684\,\text{MHz}$. (b) Power spectral density (PSD) of the fluctuations (black) and fit (gray). (c) Sketch of the parametric driving strength applied for the right and the left panel w.r.t. the instability lobe. (d) and (e): bifurcation tree of the single nonlinear resonators. Solid (dashed) lines indicate stable (unstable) DTC states for driving voltages (d) below and (e) above threshold. Dots mark bifurcation points. At each value of $f_d$, the experimental system probes one of the stable solutions, which we color code: grey (0-amplitude) and green (high-amplitude phase state). (f) Measured PSD of the fluctuations around the stationary DTC states as a function of the driving frequency $f_d$. Solid lines show the imaginary parts of the characteristic exponent $\mu$. (g) Real parts of $\mu$, corresponding to damping; diamonds represent experimental data and lines the theory results. System parameters: $Q = 295$, $f_0 = 2.6733\,\text{MHz}$, $U_{th} = 1.35\,\text{V}$, and $U_d$ is $1.3\,\text{V}$ (left panel) and $3\,\text{V}$ (right panel).

As the individual KPO modes are assumed to be decoupled, the sign of each coherent state is random. The sum over a large number of such modes will therefore approach a symmetric configuration unless the symmetries of all modes are broken with the same sign. The same argument can be applied to other systems, such as spins.

## Appendix C: Single resonator

In the following we present our fluctuation analysis for a single oscillator. Fig. 5(a) shows the measured fluctuations around a stable state (with 0-amplitude in this

case). The corresponding PSD [ Fig. 5(b)] can be used to identify the fluctuation frequency (peak frequency) and the decay rate (peak width) by using the best fit for Eq. (D3). Repeating the procedure, we can study their dependence on the driving frequency $f_d$ [ Fig. 5(f) and (g)]. We do this for driving strengths above and below threshold. Below threshold, only the stable state at 0 amplitude is available. When the driving frequency $f_d$ approaches $f_0$, the fluctuation frequency vanishes, while the decay rate $\Gamma$ splits, i.e. the fluctuations are over-damped [53]. For increased driving strength, the system hosts phase states at higher amplitudes. We analogously measure their fluctuation spectrum and find that the fluctuation frequency vanishes when the system transitions from the 0-amplitude state to the high amplitude phase state. At the same time, the decay time splits and one value approaches 0 at the transition. For a broad range away from the transition, the decay time is constant at $\Gamma = \gamma_0/2$, fixed by the dissipation coefficient $\gamma_0$.

## Appendix D: Derivation of the PSD

In the following we will derive the power spectral density (PSD) of small fluctuations around a stable state. It describes the noise-induced motion around the stable states presented it Figs. 3 and 4. The small fluctuation can be well described by linearizing the equations of motion (7) around the steady state. The noisy system is then described by

$$\dot{\delta\mathbf{Y}} = M_J \delta\mathbf{Y} + \mathbf{\Xi}, \qquad (D1)$$

where $\delta\mathbf{Y} = (\delta\alpha_1^{\mathrm{Re}}, \delta\alpha_1^{\mathrm{Im}}, \delta\alpha_2^{\mathrm{Re}}, ..., \delta\alpha_N^{\mathrm{Re}})$ with $\delta\alpha_j = \langle\delta\hat{a}_j\rangle$, $M_J$ is the Jacobian matrix of the right side of Eq. (7) evaluated at the coordinates of the selected solution $\mathbf{Y}_s = (\alpha_{1,s}^{\mathrm{Re}}, \alpha_{1,s}^{\mathrm{Im}}, ..., \alpha_{N,s}^{\mathrm{Im}})$ [53, 71, 76, 77], and the vector $\mathbf{\Xi}$ contains $2N$ uncorrelated white noise processes with PSD $\sigma^2$. The time evolution of the eigenvector components is then governed by $e^{\mu_i t}$, where the characteristic exponent $\mu_i$ is the $i^{\mathrm{th}}$ eigenvalue of the Jacobian matrix $M_J$. A state $\mathbf{Y}_s$ is stable if and only if the real parts of all eigenvalues are negative. Furthermore, the real and imaginary parts of each $\mu_i$ correspond to the typical frequency and decay rate $\Gamma$ of the fluctuations around a stable state, respectively.

Fourier transforming the Langevin equation (D1) yields

$$-i\omega_{\mathrm{FT}}\delta\mathbf{Y}(\omega_{\mathrm{FT}}) = M_J\delta\mathbf{Y}(\omega_{\mathrm{FT}}) + \mathbf{\Xi}(\omega_{\mathrm{FT}}). \qquad (D2)$$

We solve for the Fourier components $\mathbf{Y}(\omega_{\mathrm{FT}})$ and calculate the PSD of $\delta Y_i$ by $|\delta Y_i(\omega_{\mathrm{FT}})|^2$ using the PSD of the noise processes $\mathbf{\Xi}(\omega_{\mathrm{FT}})$. The same can also be done for the symmetric and anti-symmetric coordinates. In the experiment we used two coupled parametric oscillators, thus $M_J$ is a $4 \times 4$ matrix, which we can diagonalize. Using the symmetries of $M_J$ we find the eigenvalues $\mu_j$ and the corresponding eigenvectors $w_j \otimes (e_j, 1)$ with $w_j = (1, 1)$ for $j = 1, 2$ and $w_j = (1, -1)$ for $j = 3, 4$. We can now express the PSD as a function of $\mu_j$ and $e_j$ and obtain $\mathrm{PSD}_j = \mathrm{PSD}_{\alpha_j^{\mathrm{Re}}} + \mathrm{PSD}_{\alpha_j^{\mathrm{Im}}}$. Each PSD is given by

$$\mathrm{PSD}_j = \frac{\sigma^2\left(\mathrm{Im}(\mu_j)^2\left(2\mathrm{Im}(e_j)^2\mathrm{Re}(e_j)^2 + \mathrm{Im}(e_j)^4 + \left(\mathrm{Re}(e_j)^2 + 1\right)^2\right) + 2\mathrm{Im}(e_j)^2\left(\mathrm{Re}(\mu_j)^2 + \omega^2\right)\right)}{\mathrm{Im}(e_j)^2\left((\mathrm{Im}(\mu_j)^2 - \omega^2)^2 + \mathrm{Re}(\mu_j)^2(2\mathrm{Im}(\mu_j)^2 + \mathrm{Re}(\mu_j)^2 + 2\omega^2)\right)}. \qquad (D3)$$

This simple form holds true whenever the corresponding eigenvalue $\mu_j$ appears in complex conjugated pairs, but also gives very good results for real-valued $\mu_j$.

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

Phys. Rev. E **94**, 022201 (2016).