# Peer review of "The role of fluctuations in quantum and classical time crystals"

_SciPost Physics_

## Round 1 · Referee Report · Anonymous (Referee 1) · 2022-5-18

Strengths

• Timely comparative study between classical and quantum time crystals
• Contains thorough theoretical and experimental results
• Introduction is very clear and informative

Weaknesses

• The conclusions of the work to DTC beyond the model studied in the paper are questionable
• Section II. needs expanding and is not sufficiently clear
• The analysis of the full quantum regime of the model is only done for 2 modes and is therefore very limited

Report

The authors consider a series of interacting bosonic modes subject to an external drive and analyse the key features of the model using both quantum and classical treatments. The authors demonstrate that the model is capable of hosting a DTC (Discrete Time Crystal) phase with associated DTTSB (Discrete Time-Translation Symmetry Breaking) and argue, using both theoretical and experimental results, that the key features of the model can be described classically, with the quantum fluctuations playing a similar role to the classical ones.

The authors begin with an introduction of their model, moving to a rotating frame and analysing its properties when ignoring nonlinearities. In this section I believe the theoretical analysis needs to be explained in more detail. For instance, it would be helpful for the authors to discuss more thoroughly:

Change $\mathbf{1)}$ The motivation behind the model and its relation to previous works by the authors such as Phys. Rev. Lett. 123 124301 2019 and Phys. Rev. Lett. 123 173601 2019.
Change $\mathbf{2)}$ How Fig. 1b is produced for both $\gamma = 0$ and $\gamma \neq 0$, including what the actual scale for the $x$ and $y$-axes is in this Figure and the explicit relationships between the $\tilde{\Delta}_{k}$ , $\tilde{G}_{lk}$ and the parameters of the original Hamiltonian as well as the dependence of the lobe structure on these parameters. Presumably this figure is dependent on the explicit geometry used in Fig. 2a)?

The authors then move on to analyse the system for N=2 with nonlinearity. Here the authors ‘methodology is clearer and their analysis is thorough. Change $\mathbf{3)}$ The authors, however, do need to specify what the parameter κ is in the caption of Fig. 2 and what value of γ has been used here.

The authors then proceed to describe an experimental implementation of their Hamiltonian with a classical setup, stating "the resonator network can therefore be modelled by Eq. (1)". Change $\mathbf{4)}$ Here, the authors should clarify that they mean "modelled by the classical limit of Eq. (1)".

The results of their experiment then lead the authors to conclude that the qualitative features of the quantum model are captured by this classical experiment: they finish with the line that ‘there is no fundamental distinction between classical and quantum time-crystalline phases’. In my opinion, such a conclusion is far too strong given that the authors only analyse a specific quantum Hamiltonian, with specific parameters and only $N=2$ oscillators in the fully quantum regime. The dependence of the system on $N$, the oscillator geometry and the strength of the nonlinearity $V$ are not included here. The conditions for a DTC include `sufficiently long-range correlations’ (the authors words), something which cannot be analysed in the authors’ work due to the small system sizes involved. Change $\mathbf{5)}$ I would therefore suggest that the authors significantly temper their conclusions and provide some analysis, or at least discussion, of how the physics of the system changes as the size of the oscillator network increases.

To summarise, whilst the paper represents a valuable study into the role of quantum and classical fluctuations in a time-crystalline system I do not believe (for the above reasons) the paper in its current state is suitable for publication in SciPost Physics. If the authors carefully address the above concerns, then I believe the paper may be suitable for SciPost Physics.

Requested changes

See the boldfaced numbers in the report above.

---

## Round 1 · Referee Report · Anonymous (Referee 2) · 2022-6-15

Report

The article discusses periodically driven bosonic systems that may show spontaneous breaking of translational symmetry in time and formation of time crystals. Technically, the manuscript appears to be correct, but the discussion of the results in the time crystal conjunction is confusing. My more detailed comments are below.

The consequences of the existence of symmetries in classical and quantum systems are very different, but the reader has the impression that they are the same. Let me illustrate this problem with a simple example of a particle in the symmetrical double-well potential. Quantum eigenstates are superpositions of states located in both the wells, because the symmetry of the double-well potential imposes such a requirement in quantum mechanics. In the classical case, it is not a problem for a particle having a well-defined energy to be located in one of the wells. This fact does not surprise anyone, and to call it a spontaneous symmetry breaking in a similar sense to the quantum case would be misleading.

The same is true if we consider time translational symmetry. A classical particle in the presence of a time-independent potential maintains continuous time translational symmetry only when it is at rest. If a classical particle is moving, it does not meet the translational symmetry in time, but it is difficult to associate this phenomenon with the spontaneous breaking of the translational symmetry in time.

If we consider periodically driven systems, we are dealing with discrete translational symmetry in time, but (as in the above examples) its consequences in quantum and classical systems are different. A single classical particle can follow a periodic trajectory with a period, for example, twice as long as that of a periodic drive, while quantum Floquet states must evolve with the same period. The fact that spontaneous breaking of discrete time translation symmetry occurs in quantum many-body systems (in the thermodynamic limit) is a much less trivial phenomenon than the aforementioned classical periodic trajectory.

Another problem that is not clearly presented in the manuscript is the difference between time crystals in closed and dissipative systems. While in the latter it is much easier to deal with the problem of pumping energy as a result of a periodic drive, because the system can release energy to the environment, in the case of the former, breaking ergodicity is highly non-trivial. A generic closed many-body system that is periodically driven should heat up to infinite temperature (in the sense of the eigenstate thermalization hypothesis). Discrete time crystals seem to break this hypothesis. In the current version of the article, the reader may get the impression that there is no significant difference between dissipative time crystals and closed-system time crystals, which is not true.

In summary, the current version of the article is confusing and until the above-described problems are properly addressed, the article is not suitable for publication.

---

## Round 2 · Referee Report · Anonymous (Referee 1) · 2022-10-3

Report

Whilst the authors have made minor changes to the paper based on the referee comments (in the form of clarifying statements / extra descriptions) I do not believe their changes are significant enough to address the main concerns of myself and the other referee. Therefore, I do not recommend publication in Scipost Physics.

Specifically, my two main issues with the paper are:

1) The paper conveys the message that time-crystal phases are classical in character and that quantum effects are not particularly relevant. Some quotes from the most recent version of the paper: ‘demonstrating unambiguously that classical and quantum DTC share the same basic properties’ and ‘We conjecture that there is no fundamental distinction between dissipative classical and quantum time-crystalline phases in this type of system.’
2) General confusion about the rigor and level of analysis done on the Hamiltonian in Eq. (1).

My reasoning behind these issues is that:

1) There is a significant amount of literature demonstrating the unique properties of quantum time crystals (and realizing them experimentally) – see e.g. Refs [1-20]. The models considered in these references are not similar to the referee’s Hamiltonian as they often take the form of spin models which have no classical analogue. Thus, to me, the author’s Hamiltonian is not so general and their conclusions are more limited than the paper suggests.
2) The author’s analysis of the Hamiltonian for general N is confusing and unclear. The Hamiltonian in Eq. (1) is, in general, difficult to solve as it contains up to quartic terms. It is thus unclear to me:
i) In which parameter regimes (i.e. how weak does V_{j} need to be?) the results of Section II and the conclusions of the paper are valid.
ii) Which parts of the authors’ analysis on Eq. (1) are original versus which parts of their analysis stem from previous literature or the classical limit of the Hamiltonian (the authors suggest Fig. 1b was calculated from ‘the parameters of the classical system’ – I am confused as to what this means and how it relates to Eq. (1))
iii) At which point the author’s analyze the system for N > 2 in the dissipative regime. The author’s reply suggest their discussion includes the `general case of N oscillators’ yet I can only see meaningful quantitative analysis for N = 2 oscillators.

I believe that if the authors adapted the message of the paper to be more specific to the Hamiltonian in Eq. (1) and made their results clearer – then it would be appropriate for SciPost Physics Core.

---

## Round 2 · Referee Report · Anonymous (Referee 2) · 2022-10-16

Report

The authors did not make sufficient changes to the article that it would not mislead the reader. Therefore, I do not recommend the manuscript for publication. For the authors, the consequences of the existence of symmetry in quantum mechanics are the same as in classical mechanics. They do not base their theses on mathematically precise arguments, but rather on a qualitative description of the behavior of systems whose scope of applicability is difficult to assess. As with the consequences of symmetry in quantum and classical mechanics, the authors generally do not see differences between the mechanism of subharmonic evolution in dissipative and closed systems. Again, they don't base their strong statements on mathematically accurate arguments. I stand by the arguments in my first report.

---

## Round 2 · Author Response

Report 1:

We thank the referee for reading our manuscript and appreciate their feedback and questions. Below please find our reply to the points raised in the report.

1) The referee writes:

It would be helpful for the authors to discuss more thoroughly [...] the motivation behind the model and its relation to previous works by the authors such as Phys. Rev. Lett. 123 124301 2019 and Phys. Rev. Lett. 123 173601 2019.

Our reply:

Period doubling, a key feature for time crystals, is known for nonlinear parametric oscillators since a long time, see Refs. [29,30,31]. In PRL 123 124301 we showed that the period doubling effect persists for a coupled classical system and discussed different coupling strengths and their importance w.r.t. many-body effects in classical time crystals. However, with regards to time crystals, we were often confronted with the viewpoint that the classical and quantum Hamiltonian should lead to intrinsically different phenomenology. The model we present in Eq. (1) is the quantum version of the coupled equations used in PRL 123, 124301 (2019), and an extension (from one to many) of the model in PRL 123, 173601 (2019). Our goal here is really to present a unifying picture that demonstrates the analogy between all of these works.

Changes:

Following the referee’s recommendation, we highlight this relationship more clearly in the manuscript after Eq. (6).

2) The referee writes:

How Fig. 1b is produced for both $\gamma=0$ and $\gamma \neq 0$, including what the actual scale for the $x$ and $y$-axes is in this Figure and the explicit relationships between the $\tilde{\Delta}_k$ , $\tilde{G}_{lk}$ and the parameters of the original Hamiltonian as well as the dependence of the lobe structure on these parameters. Presumably this figure is dependent on the explicit geometry used in Fig. 2a)?

Our reply:

We thank the referee for pointing out this confusion. Fig 1b is a schematic representation of the instability lobes of coupled parametric oscillators. The lobe of a single mode can be calculated in a straightforward fashion from Ref. [49] and [50] for the non-dissipative and dissipative cases, respectively. Considering the case of coupled linear oscillators, the system forms normal modes subject to parametric driving. As the resonance frequencies corresponding to the normal modes differ, the instability lobes are split apart and are centered at separate resonance frequencies. Fig 1b shows an example for identical oscillators. As the normal mode frequencies depend on the coupling $J_{jk}$, this figure also depends on the geometry/the coupling matrix $J_{jk}$. We did not calculate the explicit relationships between the $\tilde{\Delta}_k$ , $\tilde{G}_{lk}$ but rather used the parameters of the classical system to obtain these figures. In Appendix A, we discuss the link between the parameters of the classical and quantum model.

**Changes: **

In the figure caption, we clarified that the Fig. 1b is a schematic phase diagram.

3. The referee writes:

The authors, however, do need to specify what the parameter $\kappa$ is in the caption of Fig. 2 and what value of $\gamma$ has been used here.

Our response:

We thank the referee for calling our attention to this typo. Indeed, $\kappa$ should be replaced by $\gamma$.

**Changes: **

We fixed the typo.

4. The referee writes:

The authors then proceed to describe an experimental implementation of their Hamiltonian with a classical setup, stating "the resonator network can therefore be modelled by Eq. (1)". Change 4) Here, the authors should clarify that they mean "modelled by the classical limit of Eq. (1)".

Changes:

We have incorporated this suggestion.

5. The referee writes:

The results of their experiment then lead the authors to conclude that the qualitative features of the quantum model are captured by this classical experiment: they finish with the line that ‘there is no fundamental distinction between classical and quantum time-crystalline phases’. In my opinion, such a conclusion is far too strong given that the authors only analyse a specific quantum Hamiltonian, with specific parameters and only $N=2$ oscillators in the fully quantum regime. The dependence of the system on $N$, the oscillator geometry and the strength of the nonlinearity $V$ are not included here. The conditions for a DTC include `sufficiently long-range correlations’ (the authors words), something which cannot be analysed in the authors’ work due to the small system sizes involved. Change 5) I would therefore suggest that the authors significantly temper their conclusions and provide some analysis, or at least discussion, of how the physics of the system changes as the size of the oscillator network increases.

Our response:

We agree that this statement seems overarching in the previous version of the paper. This statement is in fact underpinned by a combination of the following theoretical considerations:

  • Equation (1) is a very general description of an oscillating system, of which time crystals are a subset. Coupling between resonators, nonlinearity, and a parametric pump are ingredients that can be used to effectively model a wide array of physical systems.

  • Generically, realistic systems are subject to decoherence, either through internal couplings (as described at the top of page 3 of our paper) or through finite coupling to an environment. In the presence of decoherence, well-defined quantum superpositions (cat states) are lost and the long-time solutions, which are relevant for criterion (ii), can be separated into stationary points and fluctuations around these points, as we discuss in section IV.

  • These stationary points are the hot spots seen in the associated probability distribution functions of the semiclassical coherent states, see Eq. (7). These approximate the solutions of a classical system and bear no non-classical character.

  • The fluctuation of a system are dealt with after Eq. (8). These fluctuations are characterized by novel resonance frequencies and bandwidths appearing in the power spectral density. As we demonstrate experimentally, all of these features arise in an identical fashion in classical systems as well.

In summary, these considerations, borne largely out of a general theory model for $N$ resonators, led us to conjecture that the classical character of time-crystalline phases is a general property and not a consequence of the specific model we study.

With regards to the case of large N in these systems, the principal features persist, such as N normal modes leading to N instability lobes and a complex solution spacing scaling with N. Consequently, we expect that there will still be a parameter regime where our mean-field analysis holds and time crystal physics can be safely explored.

We would like to emphasize that our discussion includes the general case of $N$ oscillators, though the experimental demonstration was confined to $N=2$. We are happy to follow the referee's recommendation to formulate our conclusion in a less provocative way.

Changes:

We have replaced the relevant text by: ``We conjecture that there is no fundamental distinction between dissipative classical and quantum time-crystalline phases in this type of systems. Furthermore, we anticipate that this conclusion also holds for closed system in the prethermal regime, where time-translation symmetry can indeed be broken.''

Report 2

We thank the referee for their careful reading of our manuscript and appreciate the feedback and questions. Below please find our reply to the points raised in the report.

1. The referee writes:

The consequences of the existence of symmetries in classical and quantum systems are very different, but the reader has the impression that they are the same. Let me illustrate this problem with a simple example of a particle in the symmetrical double-well potential. Quantum eigenstates are superpositions of states located in both the wells, because the symmetry of the double-well potential imposes such a requirement in quantum mechanics. In the classical case, it is not a problem for a particle having a well-defined energy to be located in one of the wells. This fact does not surprise anyone, and to call it a spontaneous symmetry breaking in a similar sense to the quantum case would be misleading.

Our response:

We thank the Referee for this question. To clarify what we mean by spontaneous symmetry breaking, let us consider the aforementioned double-well problem. Regardless of the nature of the underlying degrees of freedom used to describe the Hamiltonian of the system (classical or quantum), a double-well potential will exhibit a symmetry between the two wells. In a quantum system, the symmetry manifests in the possibility to have quantum eigenstates that are a superposition of the particle residing in both wells. A spontaneous symmetry breaking occurs when the particle breaks the potential symmetry and collapses into one of the two wells. As stated by the referee, there can be no spontaneous symmetry breaking in a closed quantum system. The breaking of symmetry, i.e., a collapse to one of the minima, is the result of coupling the quantum particle to a dissipative environment, as was thoroughly discussed in the context of the Caldeira-Leggett model and spin-boson problems, see Annals of Physics 149, 374 (1983) and Rev. Mod. Phys. 59, 1 (1987).

It was shown that at short timescales, one can indeed see quantum coherence of the particle that allows for superposition; at intermediate timescales, the particle collapses and resides in one well in the same way a classical particle would; and at long timescales activation occurs and the particle visits the two wells stochastically with equal probability (restoring the symmetry on average). It is in this sense that we discuss the prethermal time crystal regime, wherein we find that in the semiclassical limit, the system clearly manifests the physics of TTSB, and there is no fundamental difference between the classical and quantum system.

Changes:

We now make this point clearer in the second paragraph on page 3 by adding a short version of the discussion above.

2. The referee writes:

If we consider periodically driven systems, we are dealing with discrete translational symmetry in time, but (as in the above examples) its consequences in quantum and classical systems are different. A single classical particle can follow a periodic trajectory with a period, for example, twice as long as that of a periodic drive, while quantum Floquet states must evolve with the same period. The fact that spontaneous breaking of discrete time translation symmetry occurs in quantum many-body systems (in the thermodynamic limit) is a much less trivial phenomenon than the aforementioned classical periodic trajectory.

Our response:

As shown in Phys. Rev. A 56, 4045 (1997), driven quantum oscillators are special as they have a dense quasi-energy spectra. The presence of multiple weak avoided crossings impact the convergence of any Floquet perturbative or Magnus-like expansion schemes. The presence of such dense energies makes it possible for states to manifest TTSB in states that have a higher period than that of the drive, see Phys. Rev. A 96, 052124 (2017). In this sense, as far as oscillator-based networks are concerned, one can confidently link the TTSB physics of the semi-classical regime to that of the corresponding classical system.

An alternate way of viewing this is through the prism of wave-mixing processes leading to subharmonic responses (period-doubling bifurcations). These processes are rather generic in quantum optics, circuit QED and optomechanics, regardless of the notation used to describe these systems. In line with the fact that phase transitions occur at points where maximal fluctuations appear, and where a classical description suffices, the quantum Floquet expansion must have a good classical limit that captures the same symmetry broken physics.

3. The referee writes:

Another problem that is not clearly presented in the manuscript is the difference between time crystals in closed and dissipative systems. While in the latter it is much easier to deal with the problem of pumping energy as a result of a periodic drive, because the system can release energy to the environment, in the case of the former, breaking ergodicity is highly non-trivial. A generic closed many-body system that is periodically driven should heat up to infinite temperature (in the sense of the eigenstate thermalization hypothesis). Discrete time crystals seem to break this hypothesis. In the current version of the article, the reader may get the impression that there is no significant difference between dissipative time crystals and closed-system time crystals, which is not true.

Our response:

The referee points out that we have insufficiently explained the distinction between closed and dissipative time crystals. Indeed, one of our main points was to elucidate that there is no fundamental distinction between the period-doubling mechanisms in the two cases.

To see this, first consider the dissipative case: the system is driven by the parametric drive and undergoes a period-doubling bifurcation (subharmonic response) due to the interplay between the drive and nonlinearity in the system (many-body interactions leading to wave-mixing processes). Importantly, the relevant modes that gain from the drive become strongly detuned from the drive as they grow in amplitude due to the inherent nonlinearities. This effectively decouples the mode from the drive, thereby limiting the amount of energy pumped into the system. Any excess heat is moved to the bath via dissipation. As such, the system dwells for some time in a subharmonic stationary attractor. As we show in the manuscript, the fluctuations in the system activate stochastic dynamics between the attractors, leading to an effective thermalization in the long-time limit, where the different time symmetry-broken states are sampled probablistically.

In the closed and macroscopic system: we can split the system into modes that react resonantly to the drive and modes that act as a nonresonant background. The resonant modes can undergo a period-doubling bifurcation (subharmonic response) due to the interplay between the drive and nonlinearity in the system (many-body interactions leading to wave-mixing processes). As before, these modes become detuned with respect to the drive and decouple from it due to the nonlinearity, thus limiting the amount of pumped energy. Any excess heat is then dispersed into the nonresonant background, leading at long times to infinite temperatures (ETH). Hence, here too the time symmetry-broken case appears only on prethermal time scales.

We would like to emphasize again that in bosonic nonlinear systems, the energy absorption from the drive is limited by the nonlinearity of the individual oscillators. The amount of coupling to an environment is therefore not crucial for the steady-state amplitude and there is no reason to expect the temperature to grow infinitely. This is still true for normal modes formed by a large number of oscillators coupled within a closed system as we show in section II. For instance, a large mechanical membrane formed by many atoms forms normal vibration modes. When these modes are parametrically driven, their amplitude is only affected by the damping in a minor fashion (by pushing the driving thresholds). We have now added a short clarification of these points in our manuscript.

Changes:

Following the point raised by the referee, we further clarify the role of the nonlinearity to limit the energy pumped into driven systems in the paragraph after Eq. (4).

---

## Round 3 · Referee Report · Mark Dykman (Referee 3) · 2023-2-18

Report

Yes

Attachment

---

## Round 3 · Referee Report · Anonymous (Referee 2) · 2023-2-26

Report

The revised version of the manuscript does not address my main concerns. I have no comments regarding the classical analysis presented in the article, but the quantum part and its relation to quantum discrete time crystals (DTC) are misleading. My specific criticisms are as follows: 

1/ The results of the quantum problem considered in the article correspond to a two-mode system with only a few photons (see Fig.2(c)-2(e)). Such a system has nothing to do with spontaneous symmetry breaking in quantum systems and formation of a DTC, where only in the thermodynamic limit states with broken symmetry can live infinitely long. The statement given in the introduction that "We observe that the quantum many-body system forms normal modes [fulfilling (i) and (iii)] with DTC phases that mix over time through quantum fluctuations" pertains to the case of several photons, and not to DTC, and is therefore not true. 

2/ The authors present quantum results for several photons in the system. Equation (8) correctly describes quantum fluctuations not for several photons, but in the limit of a large number of photons, i.e., in the mean-field limit. In the mean-field limit, quantum fluctuations compared to the values of \alpha_j are negligible besides close vicinity of a bifurcation point, where quantum fluctuations are large and the mean-field description breaks down. In the mean-field limit, thermal fluctuations in the quantum system can be described by equation (8), and for sufficiently high temperatures, it can be expected that they will be similar to the thermal fluctuations in the classical case. That is, it is well known that the Bogoliubov-de Gennes operator in the description of fluctuations around the mean-field state is exactly the same as the Jackobian matrix of (7), see e.g. chapters 6 and 7 in Castin, arXiv:cond-mat/0105058. Therefore, while thermal fluctuations in the quantum and classical dissipative DTCs may be similar, the connection of classical thermal fluctuations with quantum fluctuations for a small number of photons does not seem to be justified.

3/ The quantum tunneling between symmetry broken states of the two-mode system with a few photons is confused with the problem of pre-thermalization in many-body systems in the presence of many modes that couple to states corresponding to time crystals. Quantum tunneling decays exponentially quickly in the thermodynamic limit, and is not related to the pre-thermalization effect discussed in discrete time crystals (see Ref.[3]), which may be present even, and perhaps especially, in the thermodynamic limit.

4/ The system considered in the article is open. The term "dissipative time crystals" should appear in the title.

5/ The second paragraph on page 3 and Appendix B are misleading in the context of time crystals in many-body-localized (MBL) spin systems. Discrete time crystals in MBL systems are represented by pure states, and the thermodynamic limit is necessary because then they live infinitely long. Meanwhile, the authors in Appendix B consider mixed states composed of different states with broken symmetry, in which symmetry breaking and DTC will not be visible. It is like conducting many DTC experiments with pure states in the Google sycamore quantum computer Ref.[21] and then averaging the results.

6/ In the last paragraph of Sec.II, the authors write that "Opening the system to weak dissipation channels does not impact the stability diagram in a radical way." This is true for the system considered in the article, but it is not generally true for discrete time crystals in closed systems. In MBL spin systems, introducing dissipation kills MBL and DTC, see Lazarides and Moessner, PRB 95, 195135 (2017). Therefore, once again, the authors should emphasize in the title and in the article itself that they are considering a dissipative version of discrete time crystals.

7/ It is not clear to me what Hamiltonian the authors are using in (6) to find stationary states. Is it the RWA Hamiltonian (2) or the exact Hamiltonian (1)? If it is the latter, what do they mean by "stationary \rho_s," and at what time t are the results presented in Fig.2? 

In summary, I emphasize once again that I have no critical comments on the classical part, but the quantum part and its relation to quantum discrete time crystals, especially in closed systems, make the manuscript unacceptable for publication.

---

## Round 3 · Author Response

Report 1

We are grateful for the time and effort the referee invested in the second round of review. Some of the comments of the referee clearly helped us to strengthen our manuscript (see point 2). Some critique remains regarding the quantum-to-classical divide, which we believe to arise from a miscommunication on our part (see point 1). In the following reply, we attempt to clarify this matter.

The referee deems our previous revisions and explanations insufficient to warrant publication in the present form, mentioning two issues:

Point 1:

The paper conveys the message that time-crystal phases are classical in character and that quantum effects are not particularly relevant. Some quotes from the most recent version of the paper: 'demonstrating unambiguously that classical and quantum DTC share the same basic properties' and 'We conjecture that there is no fundamental distinction between dissipative classical and quantum time-crystalline phases in this type of system.'

There is a significant amount of literature demonstrating the unique properties of quantum time crystals (and realizing them experimentally) – see e.g. Refs [1-20]. The models considered in these references are not similar to the referee's Hamiltonian as they often take the form of spin models which have no classical analogue. Thus, to me, the author's Hamiltonian is not so general and their conclusions are more limited than the paper suggests.

Reply 1:

We do indeed arrive at the conclusion that quantum effects (i.e. non-classical effects) are unimportant for DTCs. We show this on the example of two coupled KPOs. Nevertheless, based on general textbook principles of quantum mechanics, it appears evident to us that the same should be true for any realistic system.

When we speak of quantum effects, we explicitly mean only effects that do not arise in classical systems, such as coherent superposition. Furthermore, our arguments only concern the long-time limit which is relevant for DTCs, and experimentally realizable systems with a finite degree of coupling to an environment. Basically, what we claim is simply that quantum coherence in an open system cannot be preserved over infinite timescales. Decoherence, in turn, leads to statistical mixing between states with different symmetries, and therefore to a restoration of the overall symmetry. A system with a restored symmetry does not count as a time crystal. In our understanding, these statements form the foundations of statistical physics and are not controversial at all.

We are aware that the time crystal community often cites the idea of a infinitely large, yet perfectly closed system composed out of perfectly uncoupled subsystems, such as spins. It is valid to speculate about such a system, and we have now revised our manuscript to mention this idea. However, we limit our argumentation to partially open systems, because each realistic system is of finite size and has a finite coupling to the outside world (especially when it is driven). The experiments that the referee cites are clearly open systems and suffer from loss of coherence over long timescales. They show time-translation symmetry breaking, which is fascinating, but do not fulfil criterion (ii).

We emphasize that it is not relevant if an experiment was conducted with spins or an oscillator. The mean-field behavior of a single spin can be described entirely with classical mechanics (see e.g. the second chapter in "Principles of Magnetic Resonance" by C. P. Slichter). Any non-classical effect arising in these spin systems, such as a quantum superposition, vanishes beyond a very short timescale.

Changes 1:

Throughout the manuscript, we emphasized that our arguments pertain to systems subject to fluctuations. Furthermore, we revised our conclusion section and formulated our claim in a more specific way to avoid misunderstandings. In appendix B, we provide the relevant quantum notation to support our discussion of coherent quantum states.

Point 2:

The author's analysis of the Hamiltonian for general N is confusing and unclear. The Hamiltonian in Eq. (1) is, in general, difficult to solve as it contains up to quartic terms. It is thus unclear to me:

i) In which parameter regimes (i.e. how weak does V_{j} need to be?) the results of Section II and the conclusions of the paper are valid.

ii) Which parts of the authors' analysis on Eq. (1) are original versus which parts of their analysis stem from previous literature or the classical limit of the Hamiltonian (the authors suggest Fig. 1b was calculated from 'the parameters of the classical system' – I am confused as to what this means and how it relates to Eq. (1))

iii) At which point the author's analyze the system for N > 2 in the dissipative regime. The author's reply suggest their discussion includes the 'general case of N oscillators' yet I can only see meaningful quantitative analysis for N = 2 oscillators.

Reply 2:

We gratefully take this opportunity to clarify these questions.

i) The theory is valid for strong coupling, while the rest of the terms are small, i.e., small detuning, small nonlinearity, parametric driving not much larger than the threshold value, and small damping. These are the usual constraints of perturbation theory, and the regime relevant for most experiments. This is the most stringent limitation beyond the fact that we only study the long-time solutions of open systems.

We added after Eq. (2) the explanation: "The RWA is equivalent to the lowest-order Floquet expansion, and relies on the fact that the corrections to the linear mode basis are nondegenerate and small [51]."

ii) There are several steps taken in the paper. We elucidate the origin and novelty of each step:

a. The standard rotating wave approximation is taken on the Eq. (1) in the most common and standard way – appearing often in the literature.

b. On top of the rotating Hamiltonian, we move to the normal mode basis – a common transformation that appears often in the literature.

c. In the normal mode picture, we describe the appearing Hamiltonian terms. We then argue, based on standard perturbation theory, which terms vanish to lowest order in the relevant parameter regimes (I-III in the manuscript) – this is also a standard approach but its application to the KPO network is new.

d. The semiclassical treatment plus study of fluctuations is a standard technique, which we apply to the KPO network.

e. The stationary solutions of the experimental system were characterized in a previous study, see Ref. [50]. Figure 2(b) is a reproduction of the theory result to serve as an overview for the reader.

iii) It is correct that we present quantitative results only for the case of N=2, because these are the results that we can compare to an experiment. However, Eqs. (2) and (6) provide a recipe for treating systems for arbitrary N. The occurrence of normal mode in the limit of a damped harmonic oscillator is obvious, and Eq. (4) describes the effects of the parametric drive in the normal-mode basis. As long as the nonlinearity is weak enough, the individual modes are essentially uncoupled and their behavior follows directly from that of a single parametric oscillator, which was treated in Ref. [46] among many other works. ---

Report 2

In our previous reply to the referee's first comment, we provided detailed arguments as to the nature of the symmetry breaking and waveform collapse in quantum systems. These arguments are based on textbook quantum mechanics and on the derivations provided in our own manuscript. The referee did not enter a scientific debate at all, accusing us instead of lacking "mathematical accuracy". The short and disparaging referee report does not contain any new information and is not constructive in a scientific environment. We hope the referee is willing to reenter a fact-based, constructive exchange with us.

For the benefit of the Referee, we added section B in the appendix to clarify the mathematical notation of quantum states. We will be happy to provide further clarification upon request.

---

## Round 4 · Referee Report · Anonymous (Referee 2) · 2023-4-15

Report
My main objections were not taken into account in the revised version of the manuscript. The article is misleading. Below, I present the necessary conditions that should be met for the article to be accepted for publication.
In dissipative DTCs, dissipation is necessary to eliminate the energy pumped by the driving, which would otherwise lead the system to an infinite temperature structureless state. Dissipative DTCs are related to dissipative structures developed by Ilya Prigogine many years ago.
The uniqueness of closed system DTCs relies on the fact that no energy drain is required from the system, yet they still exhibit periodic evolution that breaks the discrete time translation symmetry. Importantly, closed DTCs are vulnerable to dissipation, as described by Lazarides and Moessner, PRB 95, 195135 (2017). In the Google Sycamore system, dissipation is precisely responsible for the short lifetime of the DTC. Without it, DTCs would only be disrupted by quantum tunneling, which takes exponentially long in the number of q-bits. These fundamental differences between closed system DTCs and dissipative DTCs are not acknowledged by the authors.
In the manuscript, only dissipative systems are considered, and therefore this fact must be emphasized in the title to avoid confusion for readers. Additionally, the differences between closed and open system DTCs (including a reference to Lazarides and Moessner) should be presented in the article. Otherwise, the manuscript is misleading and not suitable for publication in a reputable scientific journal.
Furthermore, Appendix B, in its present form, should be removed.

---

## Round 4 · Author Response

Reply to anonymous report 3
We thank the referee for their detailed critique. The points raised by the referee underline the importance of a discussion on certain concepts in the framework of discrete time crystals (DTCs). As raised by the expert third Referee, there is confusion in the literature on nomenclature and the physics that DTCs entail, which our work rectifies. We are more than happy to summarize the main points of our manuscript once more.
Point 1:
The revised version of the manuscript does not address my main concerns. I have no comments regarding the classical analysis presented in the article, but the quantum part and its relation to quantum discrete time crystals (DTC) are misleading. My specific criticisms are as follows:
The results of the quantum problem considered in the article correspond to a two-mode system with only a few photons (see Fig.2(c)-2(e)). Such a system has nothing to do with spontaneous symmetry breaking in quantum systems and formation of a DTC, where only in the thermodynamic limit states with broken symmetry can live infinitely long. The statement given in the introduction that "We observe that the quantum many-body system forms normal modes [fulfilling (i) and (iii)] with DTC phases that mix over time through quantum fluctuations" pertains to the case of several photons, and not to DTC, and is therefore not true.
Reply 1:
We would like to start by emphasizing that we present a general theoretical treatment for a quantum system with N->infinity modes. Based on the observations in the large system, we then move to present a representative example calculated for a small photon number with two modes. However, the concept of noise-activated dynamics is generally applicable. The fact that the large-amplitude limit (or alternatively the large-network limit) exponentially increases the stability of a time crystalline state is exactly the point that we want to emphasize. In the revised manuscript, we no longer refer to this as the “classical limit” to avoid confusion. Nonetheless, it is clear that such larger amplitudes, or large (coherently coupled) networks, are much easier to fulfill in a “heavy” classical system whose amplitude substantially exceeds quantum and thermal fluctuations.
We have now removed the term ‘classical’ in the following sentences:
- in the last paragraph of section I: “Our treatment highlights that condition (ii) is much easier to fulfill in a classical system, as its large amplitudes are more resilient to fluctuations.”
- in the last paragraph on page 4, “In the limit of high normal modes' amplitudes, the influence of these quantum fluctuations rapidly decreases concomitant with a suppression of activation, cf. Appendix A.”
- in the first paragraph in section VII: “Interestingly, as massive mode populations provide resilience against fluctuation-induced activation between various attractors, systems with large amplitudes have diverging lifetimes. This condition is easy to fulfil in classical systems.”
Point 2:
The authors present quantum results for several photons in the system. Equation (8) correctly describes quantum fluctuations not for several photons, but in the limit of a large number of photons, i.e., in the mean-field limit. In the mean-field limit, quantum fluctuations compared to the values of \alpha_j are negligible besides close vicinity of a bifurcation point, where quantum fluctuations are large and the mean-field description breaks down. In the mean-field limit, thermal fluctuations in the quantum system can be described by equation (8), and for sufficiently high temperatures, it can be expected that they will be similar to the thermal fluctuations in the classical case. That is, it is well known that the Bogoliubov-de Gennes operator in the description of fluctuations around the mean-field state is exactly the same as the Jackobian matrix of (7), see e.g. chapters 6 and 7 in Castin, arXiv:cond-mat/0105058. Therefore, while thermal fluctuations in the quantum and classical dissipative DTCs may be similar, the connection of classical thermal fluctuations with quantum fluctuations for a small number of photons does not seem to be justified.
Reply 2:
Again, we fully agree. The referee echoes our point that truly non-classical phenomena differ from a mean-field approach. However, these phenomena are short-lived in the presence of fluctuations and therefore are not considered in our manuscript. In addition, as the Referee also highlights, spontaneous symmetry breaking required by DTCs manifests only in the large-amplitude/thermodynamic limit. Such massive stationary motion sustains standard quasiparticle excitations which are readily populated by quantum or thermal fluctuations. Note that our Fig. 2 in the low-photon limit is used to demonstrate the rapid convergence from quantum to the semiclassical limit in driven systems. There, even for cavities with approximately 2 photons on average, the mean-field treatment yields a good description.
Point 3:
The quantum tunneling between symmetry broken states of the two-mode system with a few photons is confused with the problem of pre-thermalization in many-body systems in the presence of many modes that couple to states corresponding to time crystals. Quantum tunneling decays exponentially quickly in the thermodynamic limit, and is not related to the pre-thermalization effect discussed in discrete time crystals (see Ref.[3]), which may be present even, and perhaps especially, in the thermodynamic limit.
Reply 3:
As we write in reply to the third referee, we agree that the terminology of ‘prethermalization’ is confusing and not necessary in the present case. We are happy to avoid it. In the revised manuscript, we exchanged the term with ‘DTC lifetime’.
The mechanism underlying spontaneous symmetry breaking in quantum systems was already discussed in our reply to the referee’s question 1 in the first round of review, see our response from September 7, 2022. To summarize, indeed, large-amplitude or thermodynamic systems become more resilient to quantum fluctuations with size.
We would like to stress that even though quantum tunneling may be suppressed in the thermodynamic limit, activation via noise remains the main reason why two-level fluctuators appear ubiquitously in physics. In many quantum engineering disciplines, the influence of spurious two-level systems on a single engineered quantum degree of freedom has been identified as a fundamental and impactful source of decoherence that proves immensely hard to mitigate.
Point 4:
The system considered in the article is open. The term "dissipative time crystals" should appear in the title.
Reply 4:
Again, this point was addressed in our reply to the referee’s question 3 in the first round of review, see our response from September 7, 2022. As we explain there and in the manuscript, dissipation as such has no important consequences in our system (apart from slightly shifting the boundaries of the phase diagram). It is the fluctuations that are crucial for driving the population of the Bogoliubov modes and for the restoration of symmetry over long timescales. We emphasize that such fluctuations are also present in closed systems without dissipation. Therefore, we prefer to not distinguish between dissipative and non-dissipative DTCs. Such a distinction leads to a misleading notion of different ‘classes’ of DTCs that, upon close scrutiny, are not fundamentally different. By using the term ‘fluctuation’ instead of ‘dissipation’, we believe our paper is clearer and more valuable to the readership.
Point 5:
The second paragraph on page 3 and Appendix B are misleading in the context of time crystals in many-body-localized (MBL) spin systems. Discrete time crystals in MBL systems are represented by pure states, and the thermodynamic limit is necessary because then they live infinitely long. Meanwhile, the authors in Appendix B consider mixed states composed of different states with broken symmetry, in which symmetry breaking and DTC will not be visible. It is like conducting many DTC experiments with pure states in the Google sycamore quantum computer Ref.[21] and then averaging the results.
Reply 5:
Limiting the discussion to coherent states without superpositions leads to a semiclassical picture without true quantum features. This fully coincides with our narrative that only the large-amplitude regime (or the large-network regime) can have exponential stability in the DTC phase. Such systems can be well described by a deterministic classical theory.
Our motivation for adding appendix B after the previous round of reports was to allow for a unifying discussion of the diverse possible quantum regimes. This appears necessary, e.g. in light of the referee’s first question in their first report. If preferred, we will be happy to remove appendix B again, as it is not important for our main message.
The Google experiment shows that a delocalized quantum state forms within the coherence time of the device, while still having time to respond to a drive with a slower rate. This is a nice demonstration of the spatial lifetime of the extended state in the device, and its ability to respond as a subharmonic Floquet state. Note, however, that under the requirements of DTCs, such a Floquet state should be very massive (thermodynamic limit), and only then fulfill criterion (ii).
In coupled bosonic cavities, we can understand that each photon in a normal mode of our system realizes such a quantum state. The large amplitude limit leads to a similar requirement as the thermodynamic limit, and the required ‘DTC-resilience’. As such, we fail to see how the Google experiment with a small system and short-lived excitations generates a more convincing proof of DTCs than a semiclassical large system that undergoes proper symmetry breaking.
Point 6:
In the last paragraph of Sec.II, the authors write that "Opening the system to weak dissipation channels does not impact the stability diagram in a radical way." This is true for the system considered in the article, but it is not generally true for discrete time crystals in closed systems. In MBL spin systems, introducing dissipation kills MBL and DTC, see Lazarides and Moessner, PRB 95, 195135 (2017). Therefore, once again, the authors should emphasize in the title and in the article itself that they are considering a dissipative version of discrete time crystals.
Reply 6:
We replaced this sentence with a specific reference to our system: “Opening our system to weak dissipation channels does not impact the stability diagram in a radical way." Regarding the terminology of dissipative DTCs, please see point 4 above.
Point 7:
It is not clear to me what Hamiltonian the authors are using in (6) to find stationary states. Is it the RWA Hamiltonian (2) or the exact Hamiltonian (1)? If it is the latter, what do they mean by "stationary \rho_s," and at what time t are the results presented in Fig.2?
Reply 7:
We analyze the stationary states of the RWA Hamiltonian Eq. (2) with dissipation. To clarify this, we now use H for the exact Hamiltonian (1) and bar{H} for the RWA Hamiltonian.
Reply to report by Prof. M. Dykman
We are grateful for the work invested by Prof. Dykman in reading our manuscript and in providing constructive criticism, which is highly appreciated. The referee’s comments indeed helped us to improve the clarity of the text.
Point 1:
I would downplay references to prethermalization. The term is used in the meaning that is somewhat different from what is frequently implied when referring to prethermalization. In particular, in paragraph 2 below Eq.(8), it is not clear what is meant by the prethermal regime. It is a cosmetic change, but worth clarifying.
Reply 1:
This is a valuable point that we are happy to implement.
Changes: we now use the terminology ‘DTC lifetime’ instead of ‘prethermalization’ to indicate the finite timescale over which a symmetry-broken state is typically stable.
Point 2:
Two paragraphs below Eq.(8) the reference to “suppression of activation” could be read as if in the classical regime activation is suppressed, whereas it is meant that activation is suppressed in the regime of large vibration amplitudes. This refers also to the statement “classical systems have diverging prethermal timescales, making them superior to their quantum counterparts as DTCs.”: what is meant here, I think, is large vibration amplitudes rather than classical fluctuations, but the text is somewhat ambiguous.
Reply 2:
We agree that our previous text was not sufficiently clear. The referee is entirely correct that it is not the classical limit in the accepted sense, but the large amplitudes that are decisive for the phenomenology we discuss.
We now avoid the term ‘classical’ in this context and explicitly refer to large amplitudes.
Point 3:
I would word more carefully the discussion of the possibility of a time crystal in the coherent regime for a closed system in the thermodynamic limit. A quantum phase transition to the corresponding state has been described in the literature. I believe the authors mean that their condition (ii) does not hold, since the concept of “stability” does not apply in a coherent regime.
Reply 3:
We fully agree with the Referee’s interpretation of our intention. The description in the literature of the closed system quantum phase transition relies on the assumption that a macroscopic quantum state may form and respond subharmonically to a drive w/o induced coupling to any other many-body state. However, AC stark shifts and power-spectral broadening (and any higher order Floquet terms) induced by the drive will effectively couple other states in the system, leading to an effective dissipation channel as well as the propensity of the closed system to heat up to infinite temperatures. Instead, we adopted an effective open system description. This is exactly the reason that we believe that the criterion (ii) can only manifest in a ‘massive’ limit, and benefits from a standard statistical physics interpretation.
We slightly rephrased the corresponding paragraph.
Point 4:
Another term to straighten out is “fluctuations forming normal modes”: the authors are talking about fluctuations of normal modes, whereas the modes themselves are formed dynamically.
Reply 4:
We now see that our formulation here was not well chosen. In the revised manuscript, we draw a distinction between the excitation modes around stationary states and the fluctuations that drive them.
“For coupled resonators, the displacements away from the stable states are themselves coupled and form normal modes (excitation quasiparticles).”
We replaced the term ‘fluctuations’ by ‘excitations’ throughout the text where it pertains to the fluctuation-driven displacement away from the stationary states.
It would be good to explain the advantageous feature of studying fluctuations of symmetric and antisymmetric combinations of the displacements from the stable states of two oscillators rather than studying such fluctuations for each oscillator separately.
We now write after Eq. (10): “The advantage of this procedure is that it isolates the spectral components of the excitations, and allows us to track the individual resonances in the presence of aliasing.”
The paper would benefit if a more careful comparison with the previous work was done. In particular, the switching between coexisting vibrational states is well-understood for a single oscillator and also for weakly coupled multiple oscillators in the quantum and classical regime, see Ref. 11 and papers cited therein. I would also compare Eq. (7) with Eq.(11) in this paper. The problem of the spectra for single oscillators was addressed in PRA 83, 052115 (2011).
As stated by the referee, Ref. [11] studies the weakly coupled oscillator network. In the mean-field limit, Eq. (11) in that paper describes a stochastic Langevin-like dynamics for the position and momenta of the individual oscillators. This equation is indeed analogous to Eq. (7) (written in terms of the expectation value of the mode annihilation operator) in our work. In paragraph 2 on page 4, where activation between solutions is discussed, we now emphasize “Such fluctuation-activated mixing was studied for individual oscillators and for weakly coupled systems, see e.g. Ref [11] and references therein. The related problem of quantum heating at zero temperature was addressed in Ref. [PRA 83, 052115 (2011)]”

---

## Editorial Decision

unknown